# Evaluating the relationship between negative foot speed and sprint performance using shoe-mounted inertial sensors

Gerard Aristizábal Pla[1,2,3], Douglas N. Martini[3], Michael V. Potter[4], Wouter Hoogkamer[3], Stephen M. Cain[2]*

**1** Department of Bioengineering, University of Pittsburgh, Pittsburgh, PA, United States of America, **2** Department of Chemical and Biomedical Engineering, West Virginia University, Morgantown, WV, United States of America, **3** Department of Kinesiology, University of Massachusetts, Amherst, MA, United States of America, **4** Department of Physics and Engineering, Francis Marion University, Florence, SC, United States of America

* stephen.cain@mail.wvu.edu

**Data Availability Statement:** The data underlying the results presented in the study are available from the Open Science Framework at the URL https://osf.io/q6ku3/.

## Abstract

Negative foot speed (i.e., the speed of the backward and downward motion of the foot relative to the body at ground contact) is a strong predictor of sprinting performance. Inertial measurement units (IMUs) are becoming a popular approach for assessing sports performance. The primary aim of this study was to use IMUs to investigate the relationship between negative foot speed and top running speed attained during a sprint on an outdoor track. The secondary aim of this study was to use IMUs to investigate the relationship between initial contact foot velocity and running speed on a stride-by-stride basis for a sprint on an outdoor track. Seventeen participants performed 80-meter track sprints while wearing a shoe-mounted IMU. The anteroposterior component, vertical component, and magnitude of the velocity of the foot at initial contact was extracted from the IMU for each stride. For the mean peak stride speed of 7.98±0.78m/s and average stride speed of 7.43±0.68m/s, the adjusted $R^2$ values were 0.27 and 0.69, 0.42 and 0.64, and 0.42 and 0.75 versus the anteroposterior, vertical, and magnitude of initial contact foot velocity, respectively. In conclusion, our findings support the common coaching tip of increasing negative foot speed to improve sprint speed. In addition, the results of this study support the use of IMUs for quantifying sprinting technique with actionable metrics.

## Introduction

High-speed running is a crucial factor influencing performance in both individual [1] and team sports [2, 3]. Thus, a significant objective in sport training is improving sprint performance. Kinetic determinants of sprint performance include the horizontal component of the ground reaction forces (GRF) or the ratio of force (i.e., the orientation of the GRF vector in the sagittal plane) [4–7]. Kinematic determinants of sprint performance include spatiotemporal

**Funding:** The author(s) received no specific funding for this work.

**Competing interests:** The authors have declared that no competing interests exist.

parameters (e.g., top speed [1], contact time [8–11], step frequency [8, 9, 11]) and leg angular velocity [9].

Traditional approaches to quantify sprint performance technique include force plates [4–7], motion capture systems [9, 11], optical measurement systems [12], high speed cameras [13], smartphone applications [13], resistance devices [14, 15], GPS [16] and pressure insoles [17]. Most of these approaches to quantifying sprint performance utilize equipment that is expensive or that have a specific and inflexible setup, making it challenging for coaches and athletes to implement measurements of sprint biomechanics in day-to-day training. Wearable inertial measurement units (IMUs) offer a cost-effective and user-friendly alternative for assessing sprinting performance technique. These wearables are small (e.g., 42x27x 11mm) and generally comprise tri-axial accelerometers and gyroscopes, that measure linear accelerations and angular rates, respectively [18].

Shoe-mounted IMUs have been used to estimate sprinting performance determinants [19–21]. Martín-Fuentes et al. [19] found that plantarflexion velocity showed the greatest association with sprint performance and that ground contact time was also associated with sprint performance, with faster sprinters running with shorter ground contact time. Shoe-mounted IMUs can also be utilized to estimate foot kinematics for sprinting with the zero-velocity update (ZUPT) method with good validity for peak sprint speeds of up to 8.00±0.88 m/s [20, 21]. In particular, IMUs and the ZUPT method provide accurate estimates of stride length and cumulative distance traveled for sprinting speeds [20].

The ZUPT method [20, 22] includes four primary steps. The stride segmentation step employs raw IMU signals to identify time points of zero velocity, indicating when the velocity of the foot is approximately zero and is relatively stationary on the ground. The rotational orientation estimation step uses an Extended Kalman filter to resolve local, sensor-fixed, IMU measurements into the global inertial frame. The translational velocity estimation step integrates linear accelerations between two consecutive zero-velocity time points and corrects for integration drift error by exploiting the assumption that the foot's velocity is zero at each zero-velocity time point. The trajectory formation step determines stride parameters (e.g., stride length) by integrating foot velocities to obtain foot position.

The ZUPT method allows for the calculation of stride velocities, stride lengths, and stride times. Stride lengths can be divided by stride times to obtain estimates of the center of mass (COM) speed, as the average COM speed during a stride must be equivalent to (or very close to) the stride speed. Calculation of stride speeds during a sprint yields a velocity curve [20, 23–25]. From the velocity curve, sprinting performance determinants such as top speed [4–7] can be extracted. Additionally, the method proposed by Samozino et al. [26] can use split times from the velocity curve as inputs to obtain kinetic determinants of sprint performance, such as the ratio of force. ZUPT implementation for fast running requires the detection of stance phases [20, 27], which enables the calculation of various spatiotemporal metrics (e.g., contact time, swing time, step frequency). IMUs can therefore be used to assess sprint performance [20]. However, these data are less actionable for coaches and sprinters. For example, faster speeds are correlated with shorter ground contact times [8–11], but this information alone may not provide specific actionable steps or interventions to improve an athlete's sprint performance. One common actionable approach by coaches for improving sprint performance focuses on increasing negative foot speed (i.e., the speed of the backward and downward motion of the foot relative to the body at ground contact) [11, 28–30]. The increase in negative foot speed referred as the "pawing" or "shipping" action of the foot, is commonly coached in athletics [28].

Several studies [11, 29, 30] have found significant correlations between the anteroposterior (AP) component of negative foot speed (i.e., the relative velocity vector at touchdown: the

velocity vector at touchdown with respect to the runner's speed), and peak running speed. Haugen et al. [11] found significant correlations between the AP component of negative foot speed and peak running speed in an indoor track for 20-meter flying sprints using a motion capture system. Murphy et al. [30] found significant correlations between the AP component of negative foot speed and peak running speed in an outdoor track for 40-60-meter sprints using high speed cameras. Clark et al. [29] found the strongest correlations between the AP component of negative foot speed and average peak running speed during the last 8 meters of a 40-meter sprint in an indoor athletic facility using a motion capture system. Therefore, the AP component of negative foot speed, measured with different devices and in different running environments, is strongly associated with sprinting performance [11, 29, 30]. Unfortunately, no studies have examined the relationship between negative foot speed and top sprinting speed with shoe-mounted IMUs.

Only one study [30] examined the relationship between the vertical (VT) component of negative foot speed (i.e., the global velocity vector at touchdown) and peak sprinting speed but did not find significant correlations. Therefore, more studies examining the VT component of negative foot speed are needed to better assess the VT component of negative foot speed as being a predictor of sprinting performance.

While previous studies [11, 29, 30] have primarily focused on the AP and VT components of negative foot speed, it is important to recognize that these components represent only part of the overall motion. By examining the complete velocity vector at touchdown, which includes both AP and VT components (i.e., the vector magnitude), we can potentially gain a more comprehensive understanding of sprinting dynamics. Unfortunately, no studies have ever tried to examine the negative foot speed vector magnitude as being a predictor of sprinting performance. Therefore, more studies examining the magnitude of the negative foot speed are needed to better assess its potential as a predictor of sprinting performance.

The main aim of this study was to use IMUs to investigate the relationship between negative foot speed and peak sprinting speed attained during an 80-meter sprint on an outdoor track. Specifically, we wanted to investigate if (1) the AP component of negative foot speed could be associated with peak sprinting speed; (2) the VT component of negative foot speed could be associated with peak sprinting speed and; (3) the magnitude of the negative foot speed vector could be associated with peak sprinting speed. Based on optical motion capture or high-speed camera studies [11, 29, 30], we hypothesized that faster peak speeds would be associated with greater AP components of negative foot speed. The secondary aim of this study was to investigate the relationship between the velocity of the foot at initial contact and stride speed across all strides during the 80-meter sprint.

## Methods

### Participants

This study involved seventeen participants (13 males, 4 females), each over the age of 18 years. Summary descriptive characteristics for all participants are listed in Table 1 [20]. Inclusion

**Table 1. Descriptive characteristics for all participants.**

| Characteristics | All participants (n = 17) |
| --- | --- |
| Sex (male/female) | 13/4 |
| Age (years) | 24.6±6.1 |
| Mass (kg) | 71.8±10.3 |
| Height (m) | 1.77±0.09 |

criteria were ability to run at a speed of 7m/s or faster and no injuries or surgeries for at least three months prior to the testing session. Exclusion criteria were orthopedic, cardiovascular, or neuromuscular conditions that could potentially affect sprint performance. All participants provided written informed consent that was approved by the Institutional Review Board of the University of Massachusetts Amherst (IRB protocol number 3143).

## Experimental protocol

The experimental protocol has been explained previously [20]. Briefly, an IMU (low-g ±16 g range, high-g ± 200 g range, ± 2000 deg/s range, sampling at 1125Hz, 16-bit resolution, mass = 9.5g; Blue Trident IMeasureU, Vicon Motion Systems Ltd, Oxford, UK) was affixed to the right shoe's medial dorsal area [27] using double-sided tape and secured with Hypafix (BSN Medical, Hamburg, Germany) tape to minimize motion artefacts (Fig 1). After a self-selected warm-up, each participant performed an 80-meter sprint at maximum effort at an outdoor track. Participants were instructed to maintain a stationary position for approximately 15 seconds prior to commencing the sprint, with the IMU aligned directly over the start line.

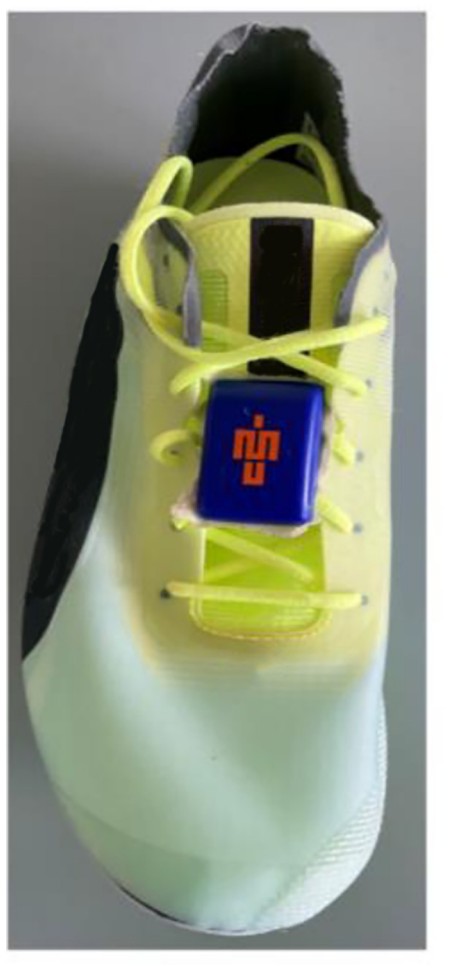
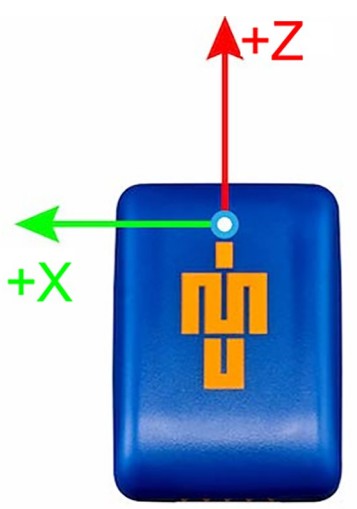
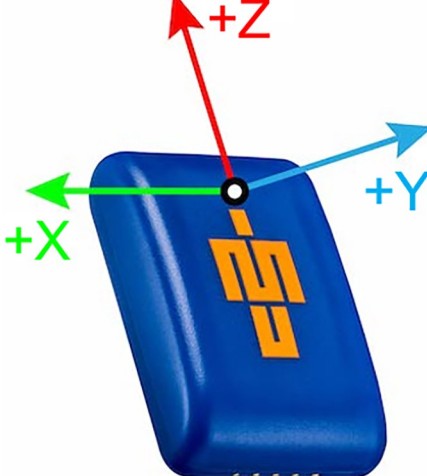

**Fig 1. Blue Trident IMeasureU attachment to the instep of the shoe.**

## IMU analysis

We analyzed the raw IMU data using customized software in Python (Python Software Foundation, Delaware, USA). To address saturation in the low-g accelerometer, the low-g and high-g accelerometers were synchronized using down sampling and cross correlation analysis. The low-g accelerometer signal was used when it was not saturated (linear accelerations smaller than ±16g) and the high-g accelerometer signal was used when the low-g accelerometer saturated [20].

The IMU-fixed frame measurements were rotated to a foot-fixed frame with axes aligned with assumed anatomical directions using a procedure similar to [31]. The average IMU-measured three-dimensional acceleration due to gravity measured during the ~15 seconds of standing still ($\overrightarrow{a}_{standing\ still}$) was used to define a foot vertical axis ($\hat{Z}_{foot}$) aligned with gravity:

$$\hat{Z}_{foot} = \frac{\overrightarrow{a}_{standing\ still}}{mag(\overrightarrow{a}_{standing\ still})} \tag{1}$$

where $mag$ denotes the vector magnitude. A foot mediolateral axis ($\hat{X}_{foot}$) was defined as an orthogonal unit vector to $\hat{Z}_{foot}$ and the IMU orthogonal unit negative Z vector ([0,0,−1]) (Fig 1):

$$\hat{X}_{foot} = \frac{[0, 0, -1] \times \hat{Z}_{foot}}{mag([0, 0, -1] \times \hat{Z}_{foot})} \tag{2}$$

where × denotes the cross product. A foot anteroposterior axis ($\hat{Y}_{foot}$) was defined as an orthogonal unit vector to $\hat{Z}_{foot}$ and $\hat{X}_{foot}$:

$$\hat{Y}_{foot} = \frac{\hat{Z}_{foot} \times \hat{X}_{foot}}{mag(\hat{Z}_{foot} \times \hat{X}_{foot})} \tag{3}$$

The resulting orthogonal vectors ($\hat{X}_{foot}, \hat{Y}_{foot}, \hat{Z}_{foot}$) were then used to define a rotation matrix that rotates the IMU measurements from the IMU frame to the foot frame.

Next, the ZUPT method was implemented and stride velocities, stride lengths and stride times were obtained [20]. Time points at foot contact, defined when the VT acceleration signal with gravity subtracted changed from negative to positive prior to maximal peaks in the acceleration magnitude, were identified (Fig 2). Note that reliable initial contact detection is difficult due to the rapid changes in running speed and initial sprint accelerations. To address this, the maximal peaks in the acceleration magnitude were selected manually and then zero crossings in the VT acceleration signal were automatically identified.

Since stride velocities are affected by yaw drift, a yaw rotation was applied to align stride velocities with the direction of running. To ensure the coordinate frame was consistently aligned with the direction of progression, we first obtained the 3D position of the foot in a global inertial reference frame where the Z axis was aligned with gravity. To account for potential yaw drift, we then defined a stride reference frame for each stride that points from foot contact at the start of the stride to foot contact at the end. The direction is then defined as follows:

$$\theta = \tan^{-1} \frac{p_{x,t=end}}{p_{y,t=end}} \tag{4}$$

where $p_{x,t=end}$ and $p_{y,t=end}$ are the displacements of the foot over time, and t = end represents

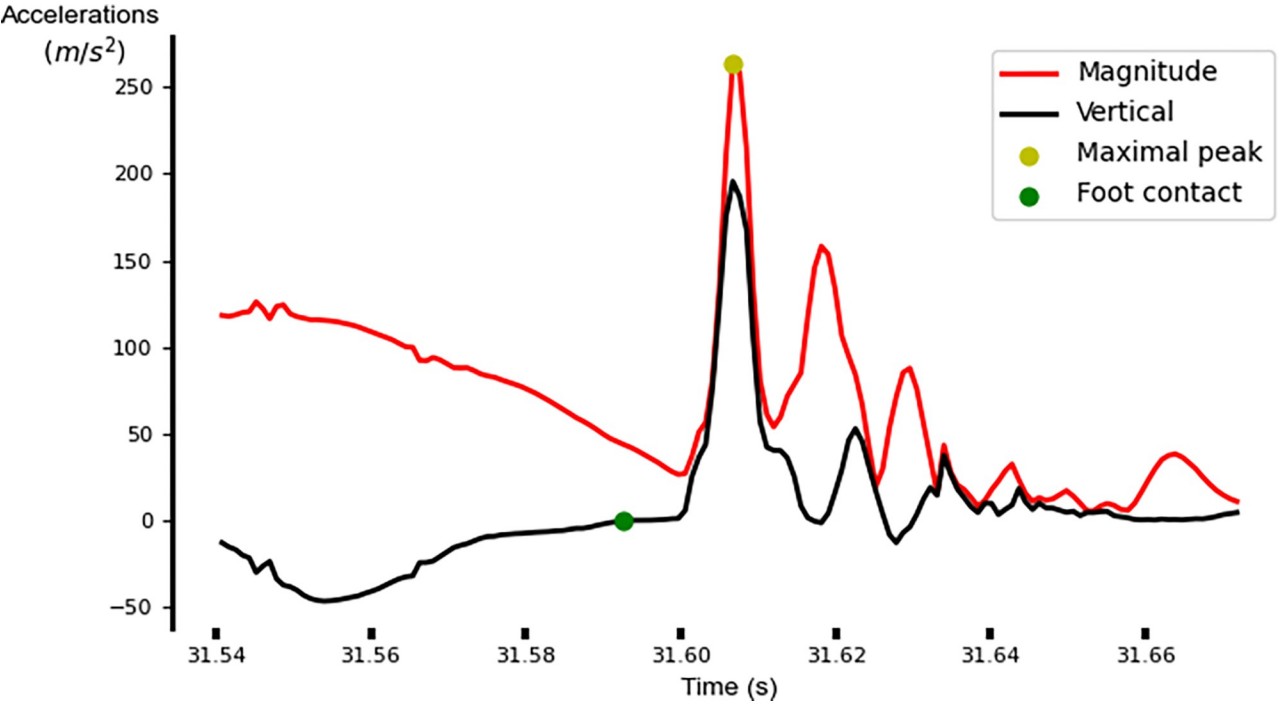

**Fig 2. Sample identification of time points at touchdown.**

the x or y displacement at the last time point (the end of the stride). This angle was then used to define a rotation matrix:

$$R_{VT} = \begin{bmatrix} \cos(\theta) & -\sin(\theta) & 0 \\ \sin(\theta) & \cos(\theta) & 0 \\ 0 & 0 & 1 \end{bmatrix} \tag{5}$$

which is applied to resolve the foot displacement and velocity in a stride-aligned reference frame. Therefore, following this rotation, for a given stride, the Y-axis and Z-axis of the stride reference frame were defined as the AP-axis (i.e., the direction of running) and the VT-axis (aligned with gravity), respectively.

Estimates of runner's stride speeds were obtained by dividing stride lengths by stride times. Finally, these speeds were subtracted from aligned foot velocities at the time points of touch-down to obtain the velocity vectors at touchdown. Peak sprinting speed was calculated for all sprints as the fastest stride speed. The velocity vector at the top speed stride was also computed. Therefore, for each participant eight values were used for the statistical analysis (i.e., top stride speed, global VT velocity vector for the top speed stride, relative AP velocity vector for the top speed stride, velocity vector magnitude at the top speed stride, stride speeds for all strides, global VT velocity vector for all strides, relative AP velocity vector for all strides and velocity vector magnitude for all strides).

## Statistical analysis

Rstudio (R Core team, Auckland, New Zealand) was used for the statistical analysis. We first confirmed that all the variables were normally distributed (i.e., $-1 <$ accepted skewness $< 1$). We then used simple linear regressions to investigate the relationship between the velocity

vector and stride speed. Stride speeds were included as dependent variables and the VT, AP and magnitude components of the velocity vector as independent variables. Correlations below 0.50 were considered weak, correlations between 0.50 and 0.80 were considered moderate and correlations above 0.80 were considered strong. Alpha level was set a priori to 0.05 for the slope of the regression and the confidence intervals.

## Results

### Relationships between initial contact foot velocity and top sprinting speed

Table 2 shows the obtained peak negative speeds along the three anatomical axes, the peak speeds, the average speeds, and the stride lengths. Peak speed occurred between 20 and 70 meters for the 80-meter sprint for all runners.

The relative AP velocity vector had a weak significant relationship to top stride speed. Faster peak speeds were achieved with greater AP components of the relative velocity vector ($R^2$ = 0.69; p = 0.02; y = -0.81–0.54x [where x denotes top stride speed]; Fig 3 and Table 3). The global VT velocity vector had a weak significant relationship to top stride speed. Faster peak speeds were achieved with greater VT components of the global velocity vector ($R^2$ = 0.42; p = 0.003; y = 0.19–0.37x [where x denotes top stride speed]; Fig 4 and Table 3). The velocity vector magnitude had a weak significant relationship to top stride speed. Faster peak speeds were achieved with greater velocity vector magnitudes ($R^2$ = 0.42; p = 0.003; y = 0.58+0.65x [where x denotes top stride speed]; Fig 5 and Table 3).

### Relationships between initial contact foot velocity and sprinting speed

Fig 6 depicts the velocity vector at touchdown with respect to distance along the three anatomical axes for a representative subject (P01) (Fig 6).

The relative AP velocity vector had a moderate significant relationship to peak stride speed. Faster speeds were achieved with greater AP components of the relative velocity vector ($R^2$ = 0.69; p<0.001; y = -0.99–0.52x [where x denotes stride speed]; Fig 7 and Table 4). The global

**Table 2. IMU-based measures for all participants.**

| ID number | Stride length (m) | IMU estimated top speed (m/s) | IMU estimated average speed (m/s) | VT negative foot velocity (m/s) | AP negative foot velocity (m/s) | Magnitude foot velocity (m/s) |
|---|---|---|---|---|---|---|
| 1 | 4.08 | 7.44 | 6.61 | -2.23 | -4.92 | 5.41 |
| 2 | 4.18 | 8.35 | 7.23 | -2.90 | -5.64 | 6.34 |
| 3 | 4.27 | 8.60 | 7.60 | -2.79 | -5.73 | 6.38 |
| 4 | 3.40 | 7.18 | 6.35 | -2.91 | -4.71 | 5.54 |
| 5 | 3.53 | 7.89 | 6.79 | -2.68 | -4.94 | 5.62 |
| 6 | 4.17 | 8.93 | 8.00 | -3.62 | -6.45 | 7.39 |
| 7 | 4.38 | 9.20 | 8.03 | -3.01 | -5.14 | 5.96 |
| 8 | 4.03 | 7.88 | 7.15 | -2.85 | -5.15 | 5.89 |
| 9 | 4.07 | 8.49 | 7.67 | -2.97 | -5.14 | 5.94 |
| 10 | 3.76 | 7.23 | 6.48 | -2.39 | -4.19 | 4.82 |
| 11 | 3.99 | 8.44 | 7.57 | -2.64 | -5.17 | 5.81 |
| 12 | 3.91 | 8.60 | 7.69 | -3.10 | -5.71 | 6.49 |
| 13 | 3.77 | 8.80 | 7.93 | -2.53 | -6.96 | 7.40 |
| 14 | 3.95 | 9.37 | 8.45 | -3.35 | -5.06 | 6.07 |
| 15 | 3.90 | 8.56 | 7.62 | -2.71 | -4.99 | 5.68 |
| 16 | 3.95 | 8.74 | 7.78 | -3.20 | -5.24 | 6.14 |
| 17 | 4.35 | 9.34 | 8.20 | -3.47 | -5.73 | 6.70 |

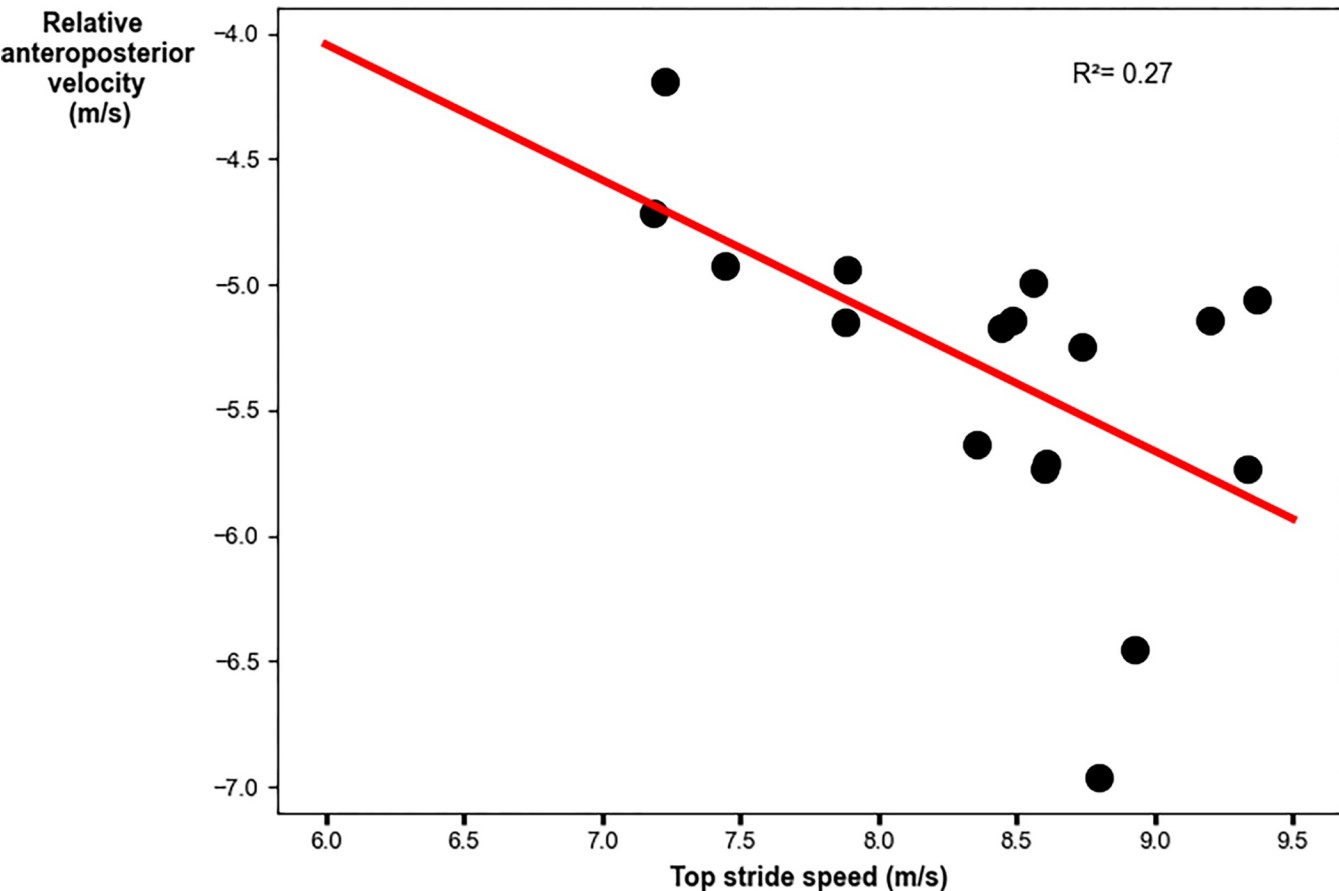

**Fig 3. Linear regression analysis for the relative anteroposterior velocity vector with peak stride speed.** Faster top speeds were achieved with greater magnitude (larger negative) anteroposterior components of the relative velocity vector.

VT velocity vector had a moderate significant relationship to stride speed. Faster speeds were achieved with greater VT components of the global velocity vector ($R^2$ = 0.64; p<0.001; y = -0.83−0.25x [where x denotes stride speed]; Fig 8 and Table 4). The velocity vector magnitude had a moderate significant relationship to stride speed. Faster peak speeds were achieved with greater velocity vector magnitudes ($R^2$ = 0.75; p<0.001; y = 1.44+0.56x [where x denotes stride speed]; Fig 9 and Table 4).

## Discussion

The main aim of this study was to investigate the relationship between negative foot speed and peak running speed attained during an 80-meter sprint. We found that the AP component, VT

**Table 3. Simple linear regression results.**

| Velocity vector component | *p* value | Adjusted $R^2$ value | Standardized β speed | Confidence intervals | Linear equation [x denotes top stride speed] |
|---|---|---|---|---|---|
| Anteroposterior | **0.018** | 0.27 | -0.57 | (-0.97, -0.11) | y = -0.81−0.54x |
| Vertical | **0.003** | 0.42 | -0.68 | (-0.59, -0.15) | y = 0.19−0.37x |
| Magnitude | **0.003** | 0.42 | 0.67 | (0.26, 1.05) | y = 0.58+0.65x |

Significant differences are highlighted in bold.

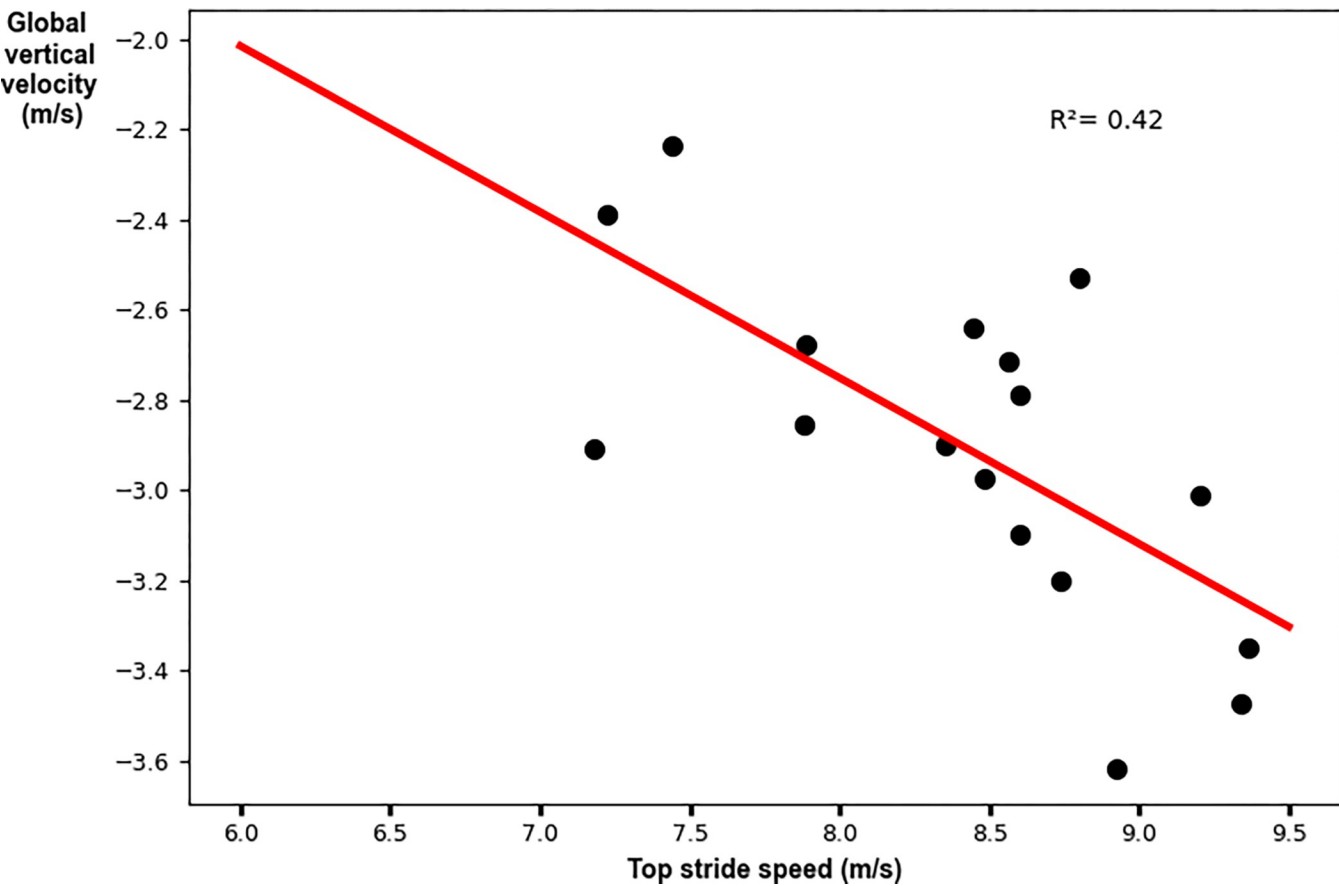

**Fig 4. Linear regression analysis for the global vertical velocity vector with peak stride speed.** Faster top speeds were achieved with greater magnitude (larger negative) vertical components of the global velocity vector.

component, and magnitude of the velocity of the foot at initial contact were weakly correlated to peak stride speed. Specifically, for peak stride speeds of 7.98±0.78m/s the adjusted $R^2$ values were 0.27, 0.42, and 0.42 for the AP component, VT component, and magnitude of negative foot speed, respectively. The secondary aim of this study was to investigate the relationship between the velocity of the foot at initial contact and stride speed across all strides during the 80-meter sprint. We found that the AP, VT and magnitude components of the velocity of the foot at initial contact were moderately correlated to stride speed. Specifically, for stride speeds of 7.43±0.68m/s the adjusted $R^2$ values were 0.69, 0.64 and 0.75 for the AP, VT and magnitude components of the velocity of the foot at initial contact, respectively. Our findings align with prior studies, confirming that the AP component of negative foot speed is an important determinant of sprinting performance [11, 29, 30]. Additionally, the results demonstrate that the VT component, as well as the magnitude of the negative foot speed vector, are important determinants of sprint performance. This kinematic parameter, encompassing all three components, is of particular interest to coaches, with an increase in these values referred to as the "pawing" or "shipping" action of the foot [28]. This interest in the "pawing" or "shipping" action and negative foot speed as key performance determinants was further underscored by interviews with over 30 coaches from various sports, who emphasized the practical impact of focusing on these metrics for sprint performance. Thus, our findings support the common coaching tip of emphasizing the "pawing" or "shipping" action to improve sprint performance.

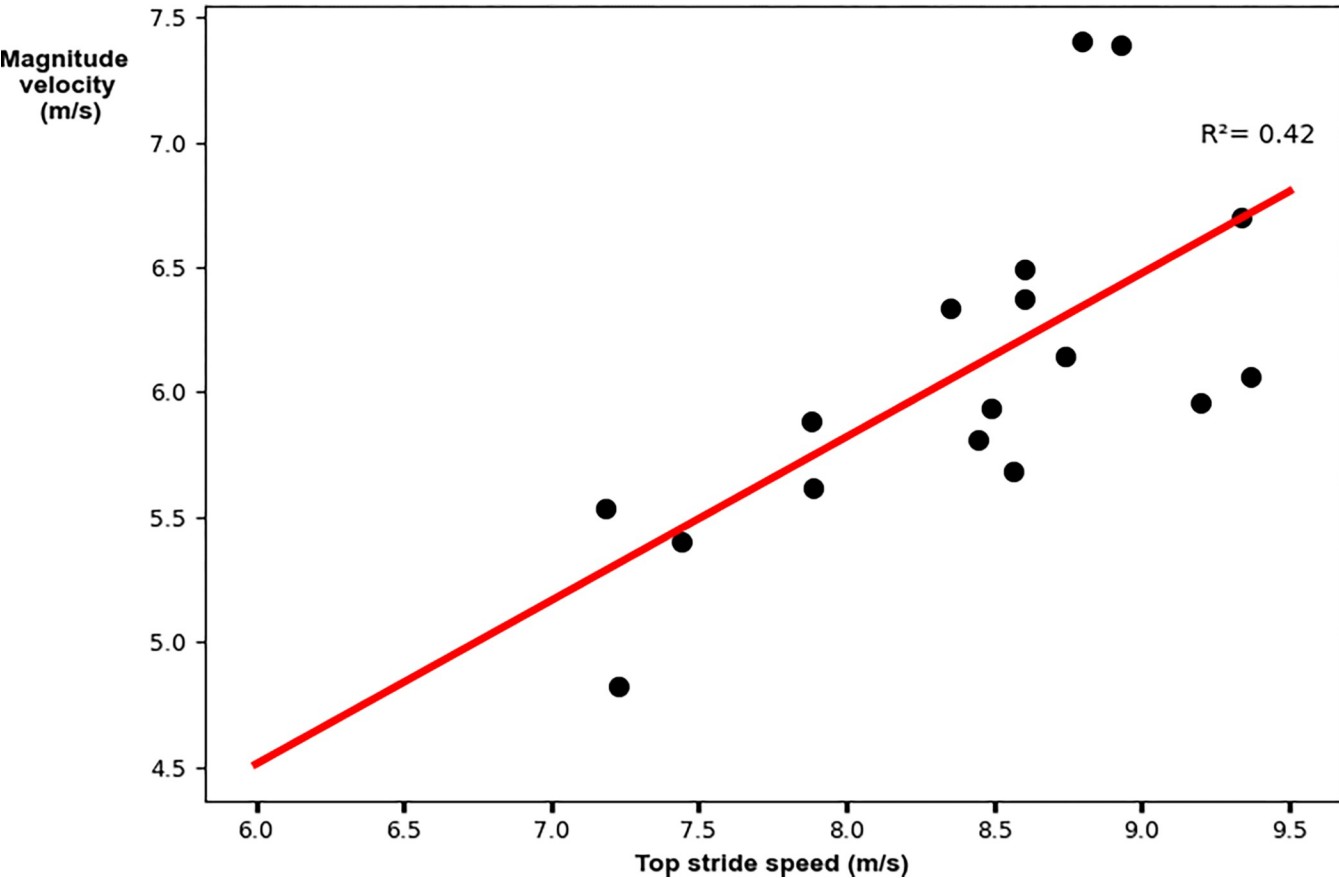

**Fig 5. Linear regression analysis for the velocity vector magnitude with peak stride speed.** Faster peak speeds were achieved with greater velocity vector magnitudes.

Our findings also build further on limited available research for assessing sprinting technique with actionable metrics using only shoe-mounted IMUs [19]. Martín-Fuentes et al. [19] found that plantarflexion velocity correlated most strongly with sprint performance and shorter ground contact times were associated with faster sprinters. We found that faster peak speeds were achieved with greater VT and AP components of negative foot speed. Therefore, we recommend the use of shoe-mounted IMUs for sprint technique evaluation. In addition, shoe-mounted IMUs have several advantages over traditional approaches to quantify sprint performance technique, such as cost-effectiveness, ease of set-up, minimal interference with the running movement, and the ability to deliver instant feedback. Furthermore, when compared to other wearable technologies like GPS devices [16] or pressure insoles [17], shoe-mounted IMUs offer superior accuracy in capturing detailed foot mechanics crucial for sprint performance. GPS units, while useful for tracking overall sprint velocity and distance, lack the accuracy needed to assess sprinting speed on a stride-by-stride basis. Pressure insoles, although informative for plantar pressure distribution, do not capture key metrics like foot speed and acceleration as comprehensively as IMUs. Therefore, our work may further increase the utility of IMUs for sprint coaching and performance evaluation over traditional approaches to quantify sprint technique.

A novel coaching approach involves athletes using foot-mounted IMUs in training to receive instant feedback on their negative foot speed. This allows them to actively focus on

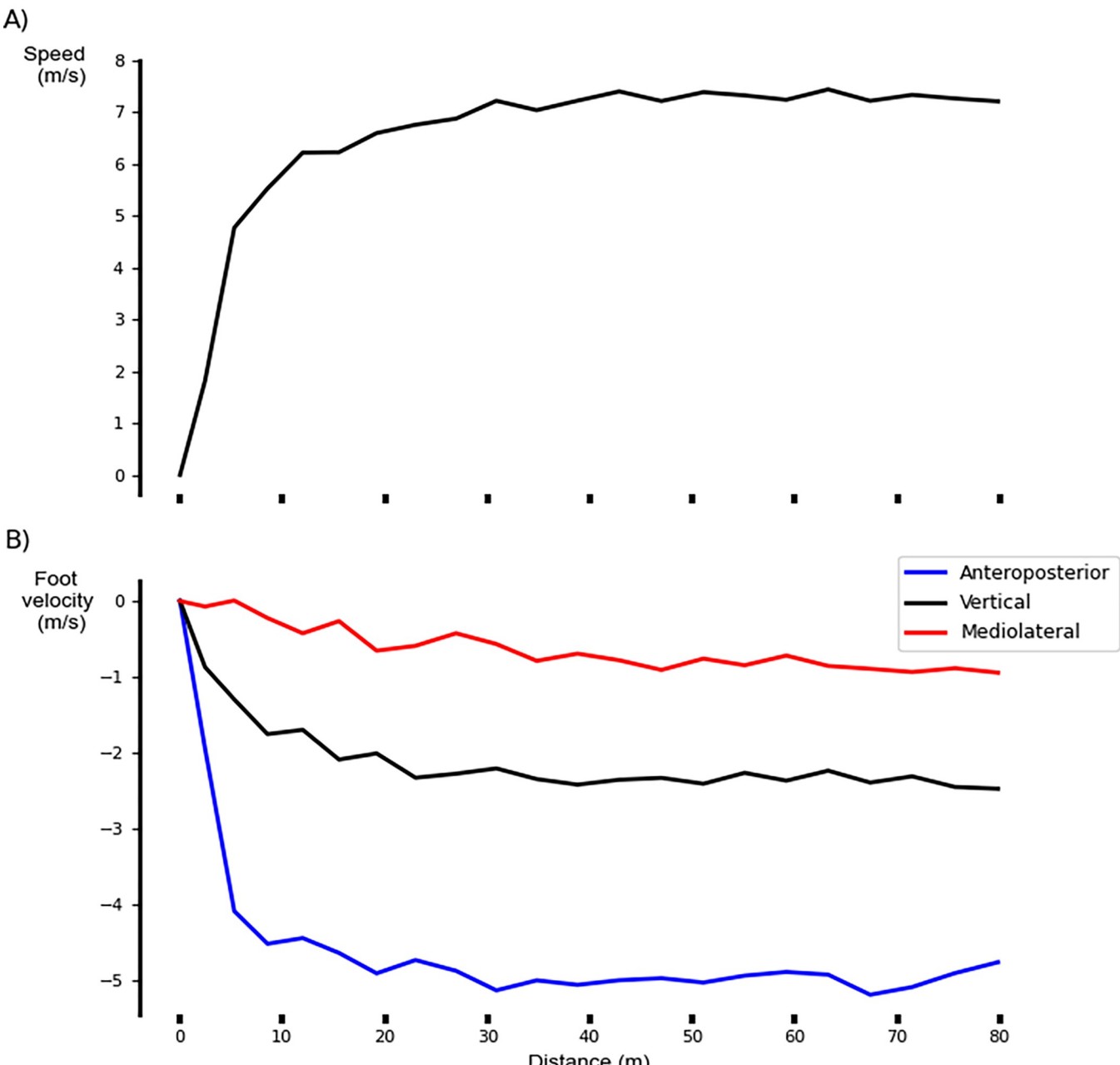

**Fig 6. IMU-derived measures for an 80-meter sprint on a stride-by-stride basis as a function of the distance traveled for a representative subject (P01).**
(A) Speed versus cumulative stride length. (B) Decomposition of foot velocity into anteroposterior (blue), vertical (black), and mediolateral (red) components.

improving this aspect of their performance, potentially leading to faster sprinting speeds. Despite these advantages, it is essential to note that while our findings indicate a significant relationship between negative foot speed and peak running speed, our study design is correlational. Thus, we cannot definitively establish causality between these variables. A "chicken and egg" scenario is possible: are faster athletes running with a more negative foot speed, or is this

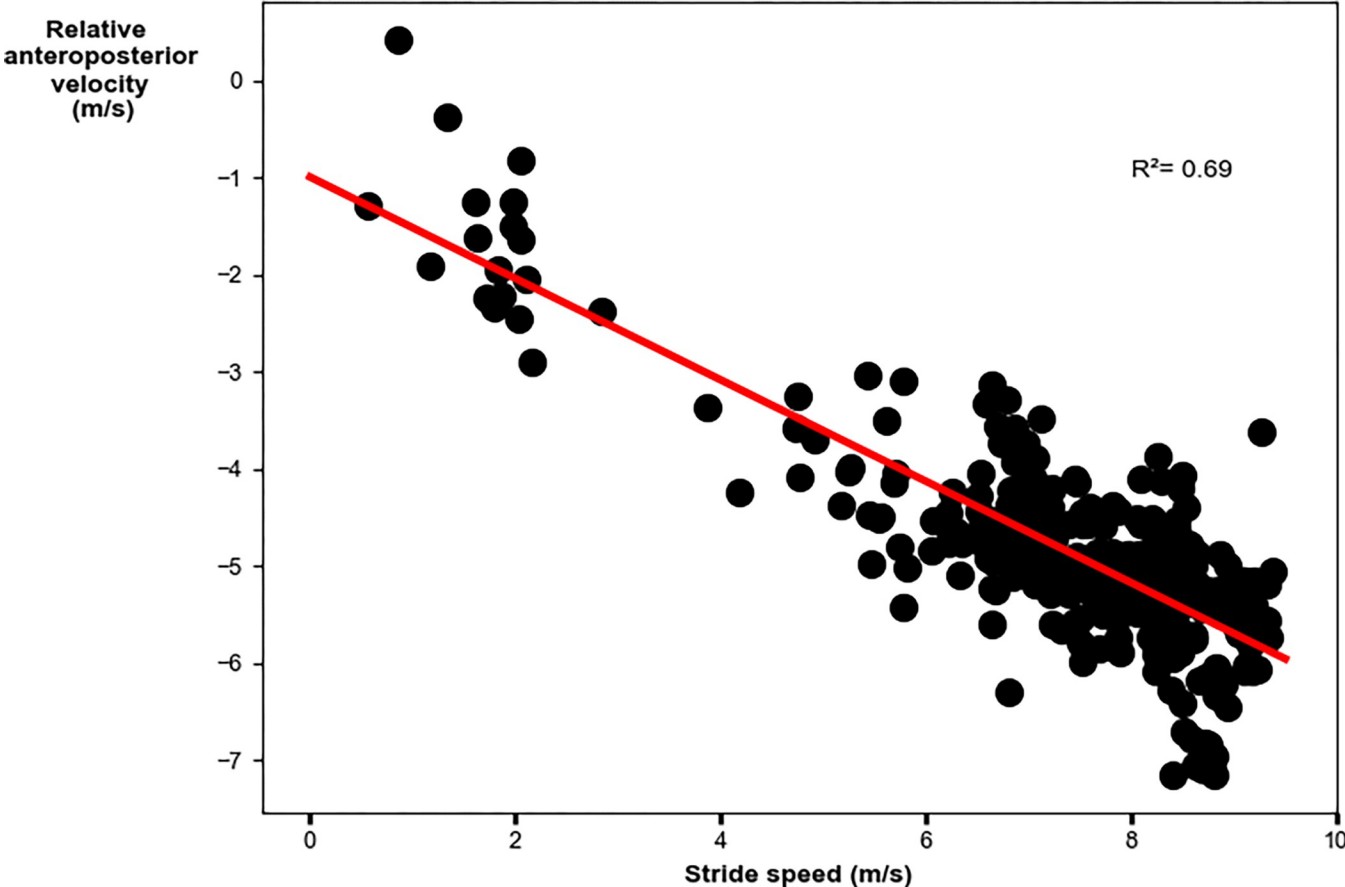

**Fig 7. Linear regression analysis for the relative anteroposterior velocity vector with stride speed.** Faster speeds were achieved with greater magnitude (larger negative) anteroposterior components of the relative velocity vector.

relationship coincidental? Coaches, however, generally believe the relationship is causal [28, 30]. They emphasize negative foot speed as a key sprinting technique to improve speed, suggesting that a more negative foot speed may reduce braking forces and allow for greater acceleration. Although further experimental research is needed to confirm a direct causal link, these insights provide a plausible explanation for why improving negative foot speed could lead to faster sprinting speeds.

Certain limitations must be acknowledged in the present study. First, IMU-based negative foot speed was not compared against the ground truth negative foot speed measured by an optical motion capture system. Previous work has validated other kinematic quantities

**Table 4. Simple linear regression results.**

| Velocity vector component | $p$ value | Adjusted $R^2$ value | Standardized β speed | Confidence intervals | Linear equation [x denotes top stride speed] |
|---|---|---|---|---|---|
| Anteroposterior | **<0.001** | 0.69 | -0.82 | (-0.56, -0.48) | y = -0.99−0.52x |
| Vertical | **<0.001** | 0.64 | -0.80 | (-0.27, -0.23) | y = -0.83−0.25x |
| Magnitude | **<0.001** | 0.75 | 0.86 | (0.52, 0.59) | y = 1.44+0.56x |

Significant differences are highlighted in bold.

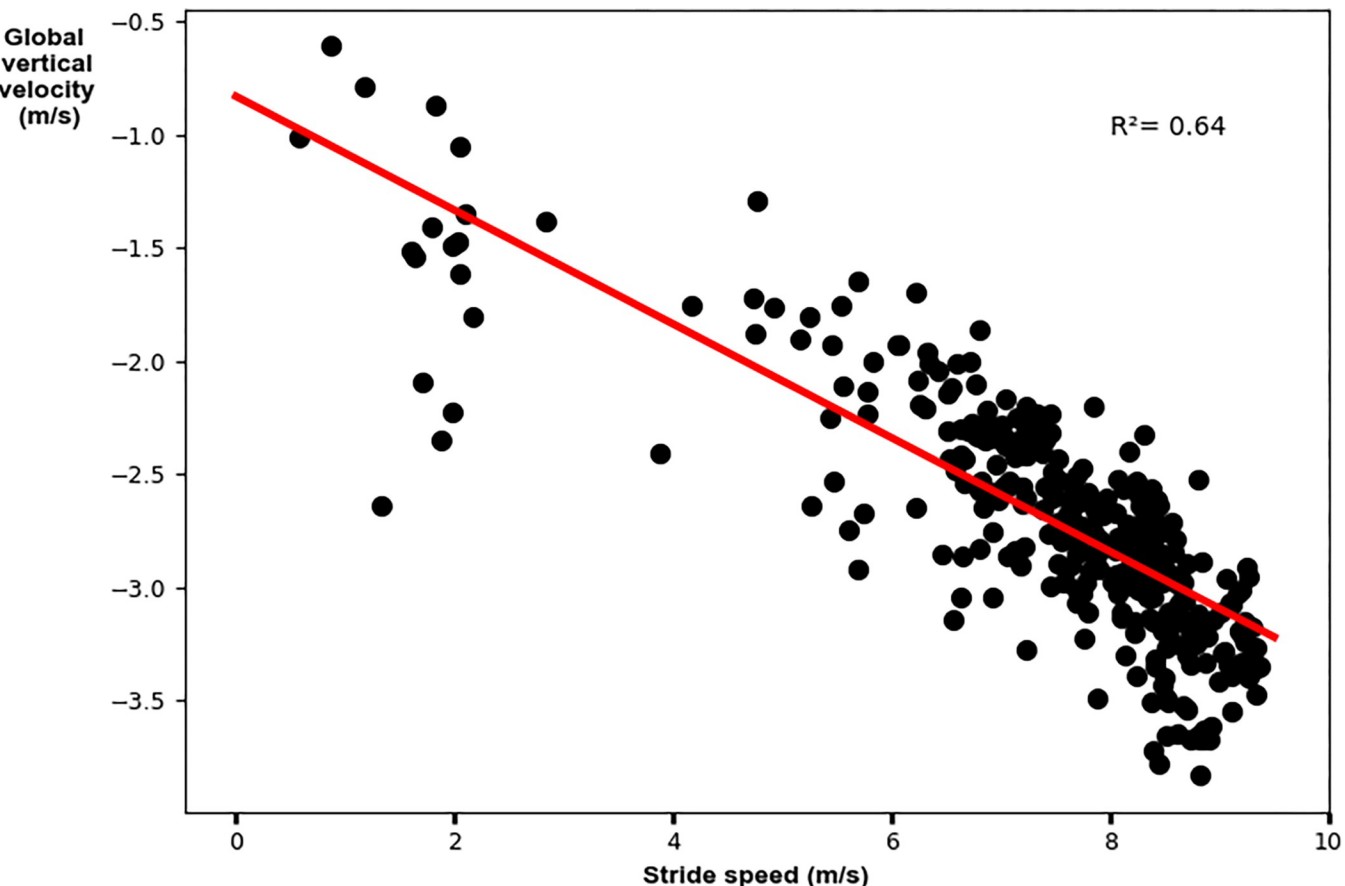

**Fig 8. Linear regression analysis for the global vertical velocity vector with stride speed.** Faster speeds were achieved with greater magnitude (larger negative) vertical components of the global velocity vector.

obtained from shoe-mounted IMUs during sprinting [20, 21, 32, 33]. Aristizábal Pla et al. [20] and de Ruiter et al. [21] validated IMU-based stride lengths with a camera-based system. Though, Aristizábal Pla et al. [20] obtained an average bias ± limits of agreement of -0.27 ±4.61% while de Ruiter et al. [21] obtained an average bias ± limits of agreement of -2.51 ±8.54%. de Ruiter et al. [32] and van den Tillaar et al. [33] validated IMU-based step detection algorithms with optical measurement systems and force plates, respectively. de Ruiter et al. [32] obtained a bias of 0.43 ms with limits of agreement of -6.40 ms while van den Tillaar et al. [33] obtained a correlation coefficient of 0.044. However, none of these studies have specifically validated IMU-based estimates of instantaneous foot speed, namely the AP, VT, and magnitude components analyzed in our study. Therefore, future research should focus on validating IMU-based negative foot speed against gold-standard optical motion capture data to ensure accuracy and reliability. Second, IMU-based peak speeds have more uncertainty than speeds measured by an optical motion capture system [20]. Fig 3 shows a significant yet relatively low correlation, particularly for the participants that ran faster than ~8.4 m/s. These differences could be due to small but expected errors in the calculation of foot speed [20, 24] or in our estimate of the body COM velocity. Studies of Olympic sprinters [34] have established that foot velocity is correlated to sprinting velocity. Therefore, measurements of additional participants who can sprint at speeds greater than 8.4 m/s are needed to determine if the results for our fastest participants are due to limitations in our measurement approach or due

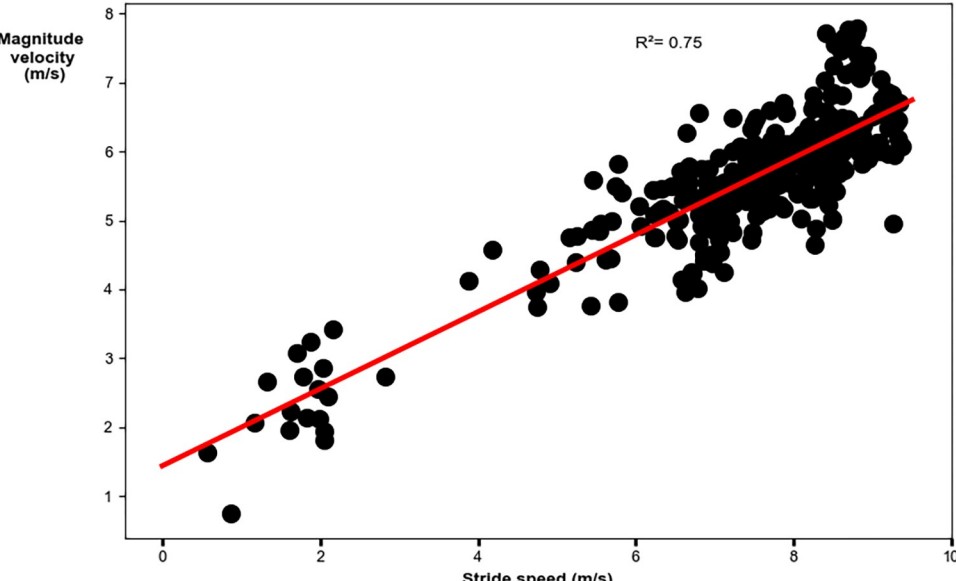

**Fig 9. Linear regression analysis for the velocity vector magnitude with stride speed.** Faster speeds were achieved with greater velocity vector magnitudes.

to individual differences in sprint technique. Third, the process of extracting metrics from raw IMU data involves complex data analysis steps, such as the rotational orientation estimation step. Users without advanced data processing skills may find these tasks challenging. Although some software platforms offer pre-processed metrics or simplified workflows, additional data analysis might still be necessary, highlighting the need for more accessible software tools to ensure usability for coaches and practitioners.

In conclusion, the results of this study support the use of IMUs for quantifying sprinting technique with actionable metrics, including the AP component, VT component, and magnitude of negative foot speed. Our results demonstrate that higher values of these components are associated with faster peak sprinting speeds, corroborating coaching practices that emphasize the "pawing" or "shipping" foot action. While IMUs offer significant advantages such as affordability, ease of setup, minimal interference with movement, and real-time feedback, several limitations must be addressed. These include the absence of comparison with optical motion capture systems and the potential challenges in data processing for users lacking advanced technical skills. Future research should focus on validating these findings against gold-standard methods and improving the accessibility of IMU-based analysis. Despite these challenges, the insights gained from this study suggest that IMUs hold substantial potential for enhancing sprint performance analysis and coaching methodologies.

## Acknowledgments

The authors express gratitude to the members of the UMILL lab who assisted in data collection. Special thank you to Dr. Alex Shorter and Dr. Loubna Baroudi for sharing their insights, Dr. Leia Stirling for her assistance in software development and ZUPT implementation, and Dr. Eric Honert for his suggestion to look at the magnitude of the negative foot speed vector.

## Author Contributions

**Conceptualization:** Gerard Aristizábal Pla, Wouter Hoogkamer, Stephen M. Cain.

**Data curation:** Gerard Aristizábal Pla.

**Formal analysis:** Gerard Aristizábal Pla, Stephen M. Cain.

**Investigation:** Gerard Aristizábal Pla, Stephen M. Cain.

**Methodology:** Gerard Aristizábal Pla, Douglas N. Martini, Michael V. Potter, Wouter Hoogkamer, Stephen M. Cain.

**Software:** Gerard Aristizábal Pla, Michael V. Potter.

**Supervision:** Wouter Hoogkamer, Stephen M. Cain.

**Writing – original draft:** Gerard Aristizábal Pla, Wouter Hoogkamer, Stephen M. Cain.

**Writing – review & editing:** Gerard Aristizábal Pla, Douglas N. Martini, Michael V. Potter, Wouter Hoogkamer, Stephen M. Cain.

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
