## [Decision Letter · Decision Letter 0]

13 Sep 2024

PONE-D-24-17404Assessing sprint technique with shoe-mounted inertial sensorsPLOS ONE

Dear Dr. Cain,

Thank you for submitting your manuscript to PLOS ONE. After careful consideration, we feel that it has merit but does not fully meet PLOS ONE’s publication criteria as it currently stands. Therefore, we invite you to submit a revised version of the manuscript that addresses the points raised during the review process.

**ACADEMIC EDITOR: **Based on the detailed comments provided by the experts, emerges the need of a deep revision of the present study. Authros should better frame the research context, strhenghten the results of their study and highlight the novelty of the work.

We look forward to receiving your revised manuscript.

Kind regards,

Andrea Tigrini, Ph.D.

Academic Editor

PLOS ONE

Journal Requirements:

Additional Editor Comments:

Based on the detailed comments provided by the experts, emerges the need of a deep revision of the present study. Authros should better frame the research context, strhenghten the results of their study and highlight the novelty of the work.

Reviewers' comments:

Reviewer's Responses to Questions

**Comments to the Author**

1. Is the manuscript technically sound, and do the data support the conclusions?

Reviewer #1: Yes

Reviewer #2: Partly

2. Has the statistical analysis been performed appropriately and rigorously? 

Reviewer #1: Yes

Reviewer #2: Yes

3. Have the authors made all data underlying the findings in their manuscript fully available?

Reviewer #1: Yes

Reviewer #2: Yes

4. Is the manuscript presented in an intelligible fashion and written in standard English?

Reviewer #1: Yes

Reviewer #2: Yes

5. Review Comments to the Author

Reviewer #1: GENERAL COMMENTS

This original study is interesting and addresses a key point for sports performance analysis: field measurements of sprint running mechanics using wearable sensors. The study has a limited scope, focusing on a single but important variable (negative foot speed at ground contact) and some limitations well acknowledged by the authors, but it brings a significant amount of actionable information for practitioners. I have specific comments I’d like the authors to address before acceptance can be recommended.

SPECIFIC COMMENTS

1. Title: maybe “sprint technique” should be replaced by something more accurate and specific of what has been actually measured/studied. Running speed and negative foot speed or kinematics?

2. The paper is overall very well written, and the introduction is well structured. I think that the “ZUPT” approach should be briefly summarized in the intro or in the methods, so that the readers understand the concept without having to read an external reference.

3. Line 78: causal/prediction terms should be used with caution in the context of a correlational design, it is preferable to say “strongly associated with sprint performance”.

4. Line 84: same comment, maybe use “would be associated with “rather than “be achieved with”.

5. In the results, it could be useful to comment on the magnitude of the correlations/relationships and not only on their significant/non-significant feature. If two correlations are significant, a moderate one and a large one should be interpreted differently.

6. Line 181: for example, these are significant but rather low correlations, so just mentioning that these variables were “significantly correlated” is misleading.

7. Line 188: I would take this relationship with caution, since the design of this study and others is mainly correlational. So, a “chicken and egg” scenario is possible: are faster athletes running with a more negative foot speed (and the correlation between negative foot speed and running speed is coincidental…) or are they running fast because of a negative foot speed? What counterarguments do we have to ascertain that it’s not a “correlation vs causation” issue, and that a more negative foot speed actually causes (at least partly) a faster running speed?

8. Line 195: it is very interesting that shoe-mounted IMUs open such possibilities, but the authors should better clarify the amount of extra data analysis work typical users will have to perform to obtain the same measurements. Is the typical user without raw data processing skills able to reproduce what is shown in this study? Some comments should be made on this point within the discussion.

9. Ket point: Figure 3 shows a significant yet pretty low correlation. The range of running speed is very wide: from 7.2 to 9.2 m/s. There is a gap without data between a group of “slow participants” maxing around 7.7 m/s and a group of faster participants running faster than 8.4 m/s. If one considers only the faster athletes (points above 8.4 m/s) this correlation seems to drop to very low, likely not significant. So, the conclusions of the study will be likely reversed when focusing only on faster athletes. This clearly changes the overall message. Could you please comment on that, and add this point to the limitations of the study (based on the correlation on the faster points only)? This is very clear in Figure 3 so any careful reader will notice that the general conclusion drawn from the full set of data does not apply to fast participants, which is a clear limitation of the approach.

Reviewer #2: This study deals with the investigation of correlations between negative foot speed and sprint performance. I suggest the acceptance of this work only if all the following points will be addressed.

Introduction

1. Please expand the description of the state-of-the-art of the instrumentations used to evaluate sprint performance both indoor and on track (e.g., pressure insoles, photo-cell systems, high-speed cameras, GPS, emg, etc.).

2. Which sprint determinants can be estimated with MIMUS? With which accuracy? For which clinical (post-injury) and sportive (performance assessment) applications?

3. Please explain better the relevance and the inertial-based methods to estimate the center of mass speed and its linkage with mentioned ZUPT method, force and power.

4. Provide a summary of the most widespread approached chosen by coaches to enhance sprint and running performance. Which is the biomechanical explanation of choosing to focus on increasing the negative foot speed?

5. Which is the novelty of this work? Which lack in the literature is filled? Which are the differences with respect to previous studies investigating the relationship between the anteroposterior foot speed and the peak running speed?

6. Provide a literature context also for vertical foot speed correlations with sprint determinants. Why did you choose to study this variable if only one study found significant correlations with peak sprint speed?

7. End the introduction with the clear explanation of the aims of the study.

Methods

8. Does the implemented method for IMU axes re-orientation guarantee that the anteroposterior axis is always aligned with the primary anatomical foot axis?

9. How the coordinate frame is rotated to be constantly aligned with the direction of progression? How was the yaw angle defined to correct the anteroposterior axis?

10. Why did you define the initial contact instants with the mentioned definition based on vertical acceleration? Mention the reference paper(s).

11. Were the performance of identification of initial contact always accurate despite great change in running speed to due initial sprint accelerations?

12. Consider to better explain the method to estimate the velocity stride-by-stride.

13. Readers may be interested in looking at correlations between the analyzed inertial-based parameters with stride-by-stride foot velocity, not only the peak speed within the sprint. Can you provide these analyses?

Results

14. At which percentage of 80-m sprint the speed peak occurred? Was it consistent among runners?

15. Were the variable distributions normal?

16. How were relative anteroposterior velocity calculated? Which is the differences with the global ones?

17. Was the method used to assess foot velocity validated? Provide information in terms of accuracy in the estimation of imu-based foot speed.

18. The results subtitle ‘Effect of sprinting speeds and body mass on stride length estimation’ is not pertinent. Add the analysis of the effect of the sprint speed (averaged within 80 m) and the body mass on negative AP and V speeds and also on stride length, which are missing.

19. Results should be improving providing a more detailed picture of the numerical findings of the work, which are now very limited. A table or diagram showing the obtained peak negative speeds along the three anatomical axis, the peak speed, the averaged speed, and the stride length would be really helpful.

20. Adding a figure depicting the foot speed with respect to time (or 0-80 m) along three anatomical axes is recommended.

Discussion

21. This section can be improved. Stress the relevance of monitoring the negative AP and V peak speed in sprinting to improve running performance. Why the correlated peak speed are important?

22. Expand the comparison with the literature in terms of assessing sprint performance with wearable technology.

23. Lines 203-204: provide a more comprehensive comparison with previous studies assessing stride length, stride velocity and/or contact time through IMUs.

24. Relevance of the work, possible scenarios of the applications of the main findings and study limitations should be stressed to provide a more complete Conclusion section.

25. Which are the novelties in suggestions in coaching approaches or in sprint performance enhancement techniques?

Title

26. Consider to rephrase the title since this study did not generally assess sprint techniques, but much more specifically analyzed the correlations between foot negative and peak speeds.

6. PLOS authors have the option to publish the peer review history of their article (what does this mean?). If published, this will include your full peer review and any attached files.

Reviewer #1: No

Reviewer #2: No

---

## [Author Response · Author response to Decision Letter 0]

6 Nov 2024

We respond to all reviewer concerns, questions, and comments in the 'Response to reviewers_vFinal.docx' file, which has been included with our submission.

---

## [Decision Letter · Decision Letter 1]

19 Nov 2024

Evaluating the relationship between negative foot speed and sprint performance using shoe-mounted inertial sensors

PONE-D-24-17404R1

Dear Dr. Cain,

We’re pleased to inform you that your manuscript has been judged scientifically suitable for publication and will be formally accepted for publication once it meets all outstanding technical requirements.

Kind regards,

Andrea Tigrini, Ph.D.

Academic Editor

PLOS ONE

Additional Editor Comments (optional):

Authors carefully revised the manuscript and update the required information. The quality of the final paper is good and I think it merits to be published.

Reviewers' comments:

Reviewer's Responses to Questions

**Comments to the Author**

1. If the authors have adequately addressed your comments raised in a previous round of review and you feel that this manuscript is now acceptable for publication, you may indicate that here to bypass the “Comments to the Author” section, enter your conflict of interest statement in the “Confidential to Editor” section, and submit your "Accept" recommendation.

Reviewer #1: All comments have been addressed

2. Is the manuscript technically sound, and do the data support the conclusions?

Reviewer #1: Yes

3. Has the statistical analysis been performed appropriately and rigorously? 

Reviewer #1: Yes

4. Have the authors made all data underlying the findings in their manuscript fully available?

Reviewer #1: Yes

5. Is the manuscript presented in an intelligible fashion and written in standard English?

Reviewer #1: Yes

6. Review Comments to the Author

Reviewer #1: Thank you for addressing all my points in a satisfactory way, the paper has improved clearly, congratulations on a very interesting work

7. PLOS authors have the option to publish the peer review history of their article (what does this mean?). If published, this will include your full peer review and any attached files.

Reviewer #1: No

---

## [Editor Report · Acceptance letter]

10 Dec 2024

PONE-D-24-17404R1 

PLOS ONE

Dear Dr. Cain, 

I'm pleased to inform you that your manuscript has been deemed suitable for publication in PLOS ONE. Congratulations! Your manuscript is now being handed over to our production team.

Kind regards, 

on behalf of

Dr. Andrea Tigrini 

Academic Editor

PLOS ONE